Adaptive changes in chlorophyll content and photosynthetic features to low light in Physocarpus amurensis Maxim and Physocarpus opulifolius “Diabolo”

Zhang Huihui 1
Zhong Haixiu 2
Wang JIfeng 2
Sui Xin 2
Xu Nan 2 xunan0451@126.com
1 College of Resources and Environment, Northeast Agricultural University , Haerbin, Heilongjiang Province , China
2 Natural Resources and Ecology Institute, Heilongjiang Academy of Sciences, Ecology , Haerbin, Heilongjiang Province , China
Sun Xiaolin
Electronic publication date: 2016 Jun 23
Publication date: 2016
Volume: 4
Electronic Location ID: e2125
Received 2016 Mar 24; Accepted 2016 May 23
Copyright: © 2016 Zhang et al.
Copyright year: 2016
Copyright holder: Zhang et al.
License: This is an open access article distributed under the terms of the Creative Commons Attribution License, which permits unrestricted use, distribution, reproduction and adaptation in any medium and for any purpose provided that it is properly attributed. For attribution, the original author(s), title, publication source (PeerJ) and either DOI or URL of the article must be cited.
License URL: https://creativecommons.org/licenses/by/4.0/

Keywords: Physocarpus amurensis Maxim, Physocarpus opulifolius “Diabolo”, OJIP curve, PS II function

Funding: “Twelfth Five-Year” National Science and Technology Support Program of China 2011BAD08B02-3 Major Project for the Heilongjiang Province Science and Technology Program GZ13B004 Project for the Heilongjiang Province People’s Government WB13B104 National Natural Science Fund 3150020596 and 31500323 Funding was provided by “Twelfth Five-Year” National Science and Technology Support Program of China: 2011BAD08B02-3, Major Project for the Heilongjiang Province Science and Technology Program: GZ13B004, Project for the Heilongjiang Province People’s Government: WB13B104, and the National Natural Science Fund: 3150020596 and 31500323. The funders had no role in study design, data collection and analysis, decision to publish, or preparation of the manuscript.

==============================
The present study aims to investigate the differences in leaf pigment content and the photosynthetic characteristics under natural and low light intensities between the Chinese native Physocarpus amurensis Maxim and the imported Physocarpus opulifolius “Diabolo” from North America. We aim to discuss the responses and the adaptive mechanism of these two cultivars of Physocarpus to a low light environment. The results show that the specific leaf area (SLA) and the chlorophyll content were significantly increased in the leaves of both Physocarpus cultivars in response to a low light intensity, and the SLA and chlorophyll content were higher in the leaves of low light-treated P. opulifolius “Diabolo” compared with the leaves of low light-treated P. amurensis Maxim. Moreover, the content of anthocyanin was markedly reduced in the leaves of P. opulifolius “Diabolo” under low light intensity, which allowed for a greater capacity of photon capture under the low light condition. Under natural light, the photosynthetic carbon assimilation capacity was greater in the leaves of P. amurensis Maxim compared with the leaves of P. opulifolius “Diabolo” that were rich with anthocyanin. However, in response to low light, AQY, Pmax, LCP and LSP decreased to a lesser extent in the leaves of P. opulifolius “Diabolo” compared with the leaves of P. amurensis Maxim. These results suggest that P. opulifolius “Diabolo” exhibits a greater ability in adaption to low light, and it is probably related to the relatively higher chlorophyll content and the smaller SLA in the leaves of P. opulifolius “Diabolo.” In addition, the low light intensity resulted in a reduced photochemical activity of photosystem (PS) II in the leaves of both Physocarpus, as evidenced by increased values of the relative variable fluorescence at point J and point I on the OJIP curve. This result suggests that the electron acceptor in PS II was the major responsive site to the low light stress in the leaves of both Physocarpus cultivars, and that the low light intensity significantly inhibited electron transfer on the acceptor side of PS II and reduced the activity of the oxygen-evolving complex (OEC) in the leaves of both Physocarpus cultivars. The PS II function in P. opulifolius “Diabolo” was higher than that in P. amurensis Maxim in response to low light. Under low light, the composition of photosynthetic pigments was altered in the leaves of P. opulifolius “Diabolo” in order to maintain a relatively high activity of primary photochemical reactions, and this is the basis of the greater photosynthetic carbon assimilation capacity and one of the main reasons for the better shade-tolerance in P. opulifolius “Diabolo.”

Introduction

Physocarpus amurensis Maxim is a deciduous shrub belonging to the family Rosaceae, and is an endangered plant species in China (Yin, Shen & Lan, 2010). Physocarpus opulifolius “Diabolo,” which was recently imported from North America into China, is a colorful ornamental species of within the genus Physocarpus (Yin, Shen & Lan, 2010). Physocarpus plants are shrubs with elegant bell-shaped flowers with dense inflorescences that form red fruits in early autumn, which gives Physocarpus plants high ornamental value. In addition, Physocarpus exhibits a high resistance to cold, and both P. amurensis Maxim and P. opulifolius “Diabolo” can be planted outdoors in North China (Yin, Shen & Lan, 2010; Liu & Yu, 2011). The abundant anthocyanin in the leaves of P. opulifolius “Diabolo” gives them a rich purple color, making them an important plant contributing to the variety of landscaping. Moreover, the bark of Physocarpus is enriched in triterpenoid compounds that exhibit anti-tumor properties, thus, Physocarpus also has a great economic value (Liu & Yu, 2011).

Light is an essential ecological factor that facilitates photosynthesis and influences growth, morphogenesis, and survival of plants (Zhang et al., 2014). Urban garden landscaping often draws from a collection of tree, shrub and grass. Because of the requirement of greening layers, and the increasing number of high-rise buildings in the modern city, many garden shrubs and grass-areas are overshadowed. The light condition can affect the morphology of the plant by reducing of root shoot ratio and the specific leaf weight, enlarging specific area, and changing leaf pigment. For instance, low-light environments interfere with normal photosynthetic activity in the plants by affecting the synthesis of photosynthetic pigments and the ratio of various pigments (Zhao, Hao & Tao, 2012). Color-leaf plants in particular are sensitive to light, as the synthesis of anthocyanin mostly depends on the plant receiving light, therefore shading can directly compromise the synthesis of anthocyanin in some plants (Michal, 2009; Meir et al., 2010; Chen, Li & Cheng, 2008).

In addition, light can influence the function of the photosynthetic apparatus in the leaves by altering the absorption and utilization of optical energy and electron transport. Photosystem (PS) II is one of the most important protein complexes in the photosynthetic apparatus of higher plants. As a fast and non-invasive probe for PS II function, fast chlorophyll florescence dynamics can detect absorption and utilization of light, distribution of excessive energy and electron transfer by PS II during the process of chlorophyll fluorescence quenching (Jiang et al., 2006; Xia, Li & Zou, 2004). Moreover, the structure and function of PS II in plant leaves can be determined by the JIP-test (Xia, Li & Zou, 2004). The studies on the light adaptive mechanism in the endangered and imported plant species can provide useful information for the conservation of the species and for a reasonable collection of imported plants (Kim et al., 2011; Perrin & Mitchell, 2013). In this study, we measured the content of photosynthetic pigments, the photosynthetic gas exchange parameters and the function of PS II in the leaves of P. amurensis Maxim and P. opulifolius “Diabolo” under natural light or shade conditions. By analyzing the changes in these parameters, we identified the effects of low light on the photosynthesis in P. amurensis Maxim and P. opulifolius “Diabolo.” Our study provides some fundamental data for a reasonable collocation of Physocarpus plants in landscaping.

Materials and Methods

Plant material

Three-year old cutting (delete) seedlings of P. amurensis Maxim and P. opulifolius “Diabolo” were provided by the Forest Botanic Garden of Heilongjiang province. In the spring of 2012, the live cutting seedlings of P. amurensis Maxim and P. opulifolius “Diabolo” were planted in peat soil in individual plastic pots with an upper diameter of 28 cm, a lower diameter of 15 cm and a height of 20 cm. The seedlings were planted outdoor in natural condition, with regular watering and weeding. The seedlings were thoroughly watered before winter, and grew outdoor through the winter. In early 2014, the seedlings were about 0.5 m in height and with 8–10 branches. After thawing in the spring of 2014, routine watering and weeding were resumed, and the two cultivars of Physocarpus in the vigorous growth stage was subjected to experiment in June 2014.

Experimental design

P. amurensis Maxim and P. opulifolius “Diabolo” seedlings (10 each) of similar size were selected. Five cuttings of each species were cultivated indoors under an artificial low light (Microwave sulfur lamp-MSL1000N1, NingBo, China) with a lighting intensity of 100 μmol · m−2 · s−1 (Measured by LIGHTSCOUT, Spectrum, USA). The other seedlings remained under an outdoor natural light with a lighting intensity between 1,000 μmol · m−2 · s−1 and 1,500 μmol · m−2 · s−1 at noontime. Plants under low light or natural light were watered in the same manner, and fertilizers were not applied. In later 45 d, the indoor seedlings of both species showed signs of adaptation to low light. Measurements for leaf pigments, light responsive curve and chlorophyll fluorescence parameter were conducted at this 45 d stage for both species.

Carotenoids and anthocyanin contents

The concentrations of chlorophyll a (Chla), chlorophyll b (Chlb) and carotenoids (Car) were determined on fresh fully expanded leaves of P. amurensis Maxim and P. opulifolius “Diabolo.” A fresh leaf sample without main vein was sliced and incubated in pigment extraction solution containing acetone, anhydrous ethanol and distilled water (4.5:4.5:1, V:V:V). Contents of Chla, Chlb and Car were calculated according to Tang, Huang & Wang (2004) and Pirie & Mullins (1976). Chla/Chlb and Car/Chl also were calculated. Anthocyanin (Ant) concentrations were determined using 0.1 mmol · l−1 HCl extraction solution. Ant concentrations and Ant/Chl were calculated with the equations proposed by Pirie and Mullins.

Measurement of photosynthetic rate

Photosynthetic rate–Photosynthetic photon flux density (Pn–PFD) response curves were made at leaf chamber temperature of 30 °C, and at 350 μmol · m−2 · s−1 CO2 with a portable photosynthetic system (CIRAS-3, PP systems, UK). PFD was fixed every 10 min in a sequence of 2,000, 1,600, 1,200, 800, 600, 400, 300, 200, 150, 100, 0 μmol · m−2 · s−1. Light intensity, CO2 concentration and leaf chamber temperature were controlled by automatic control device of the CIRAS-3 photosynthetic system. Photosynthetic rate measured at two O2 concentrations (21% O2 + 350 μmol · m−2 · s−1 CO2 and 2% O2 + 350 μmol · m−2 · s−1 CO2) under 1,200 μmol · m−2 · s−1 PFD was used to calculate photorespiration.

Measurement of chlorophyll

Chlorophyll a fluorescence transient was measured with a Handy-PEA fluorometer (Hansatech, UK). Seedlings leaf having been dark adapted for 1 h before heat treatment, all the leaves were immediately exposed to a saturating light pulse (3,000 μmol · m−2 · s−1 PFD) for 2 s after heat treatment in the dark at different times. Each transient obtained from the dark-adapted samples was analyzed according to the JIP-test (Zhang et al., 2012; Strasser, Srivastava & Govindjee, 1995) by utilizing the following original data: (1) the fluorescence intensity at 20 ms (Fo, when all RCs of PSII are open); (2) the maximum fluorescence intensity (FM, when all RCs of PSII are closed) and (3) the fluorescence intensities at 300 ms (K-step), 2 ms (J-step) and 30 ms (I-step). The maximum quantum yield of PSII photochemistry (Fv/Fm) was calculated as: Fv/Fm = (Fm − Fo)/Fm, in this study, Fm = FP. The relative variable fluorescence intensity at J-step (VJ) and I-step (VI) were calculated as: Vt = (Ft − Fo)/(Fm − Fo). VK and VL were the relative variable fluorescence on the VO-J and VO-K point at 0.3 and 0.15 ms.

According to the JIP-test (Zhang et al., 2012; Strasser, Srivastava & Govindjee, 1995) could obtain such as that maximum quantum yield of PSII photochemistry (Fv/Fm), Performance index on absorption basis (PIABS), Probability that a trapped exciton moves an electron into the electron transport chain beyond QA- (at t = 0) (Ψo), Quantum yield for electron transport (at t = 0) (φEo), quantum yield of absorption flux to dissipated energy (φDo), Absorption flux per RC (ABS/RC), Trapped energy flux per RC (at t = 0) (TRo/RC), Electron transport flux per RC (at t = 0) (ETo/RC) and Dissipated energy flux per RC (at t = 0) (DIo/RC).

Determination of specific leaf area

Five fully expanded leaves were randomly selected from each plant. The surface area of each leaf (S (cm2)) was measured. Then, the leaf was heated (105 °C, 30 min) and dried (60 °C, 30 h) until it reached a constant weight, then its biomass (M (g)) was weighed. The surface area (S) was divided by the mass (M) to obtain the SLA (specific leaf area).

Data analysis

Each experiment was repeated three times. Data represent mean ± SE. Statistical analysis was carried out with Excel and SPSS statistical software. One-way ANOVA and LSD were used to analyze all data. Differences were considered significant if p ≤ 0.05 and very differences were considered significant if p ≤ 0.01.

Results and Analysis

Chlorophyll content in the leaves of Physocarpus under different light intensities

In order to understand how exposure to lower light may affect P. amurensis Maxim and P. opulifolius “Diabolo,” we placed both plant species under natural light and low-light conditions (Fig. 1). The leaves of P. amurensis Maxim under low light became more greenish compared with the same species under natural light. The leaves of P. opulifolius “Diabolo” under low light turned green from purple, and new leaves were generally green (Fig. 1). To quantify these differences in terms of chlorophyll content, we measured the Chla and Chlb content in the leaves. We did not observe a significant difference in the contents of Chla in the leaves of P. amurensis Maxim and P. opulifolius “Diabolo” under natural light (Figs. 2A and 2B). However, under natural light conditions, P. opulifolius “Diabolo” had 56.27% (p < 0.01) higher Chlb and 20.66% higher Chl(a + b) (p < 0.01) compared to P. amurensis Maxim. However, the ratio of Chla to Chlb (Chla/b) in the leaves of P. amurensis Maxim was 48.96% (p < 0.05) higher than P. opulifolius “Diabolo” (Figs. 2C and 2D).

Figure 1 Color changes in the leaves of Physocarpus amurensis Maxim and Physocarpus opulifolius “Diabolo” under natural and low-light intensities.

Figure 2 Content of Chla (A), Content of Chlb (B), Chl(a + b) (C) and Chla/b (D) in the leaves of P. amurensis Maxim and P. opulifolius “Diabolo” under natural and low light intensities.

Bar graphs depict mean ± SE, values followed by different small letters mean significant difference (p < 0.05), values followed by different capital letters mean very significant difference (p < 0.01).

The chlorophyll content was higher in the leaves of P. amurensis Maxim and P. opulifolius “Diabolo” under low light compared to the natural light conditions, but the Chla/b ratio was significantly decreased. The content of Chla in the leaves of P. amurensis Maxim increased by 7.01% (p < 0.05) under low light conditions, and the content of Chla in P. opulifolius “Diabolo” was not different between the two conditions. The contents of Chlb and Chl(a + b) were significantly elevated in P. amurensis Maxim and P. opulifolius “Diabolo” under low light conditions (p < 0.01). The Chla/b ratio was reduced by 54.41% (p < 0.01) in the leaves of P. amurensis Maxim and 38.35% (p < 0.05) in P. opulifolius “Diabolo” under low light conditions (Figs. 2C and 2D).

Contents of carotenoids and anthocyanin in the leaves of Physocarpus under different light intensities

Based on the change in color we observed in the leaves, we measured the levels of carotenoids and anthocyanin in the leaves. The contents of carotenoids (Car) in the leaves of P. amurensis Maxim and P. opulifolius “Diabolo” were slightly increased but not statistically significant in low light compared to natural light conditions (Fig. 3A). The leaf Car contents were not significantly different between P. amurensis Maxim and P. opulifolius “Diabolo” under natural light or low light conditions. The ratio of Car to Chl (Car/Chl) in P. amurensis Maxim and P. opulifolius “Diabolo” were lower in low light conditions, but also not statistically significant (Fig. 3B). Under natural light conditions, the anthocyanin (Ant) content of P. opulifolius “Diabolo” was 3.52 times higher than P. amurensis Maxim. Ant content in P. opulifolius “Diabolo” under low light condition was 65.33% (p < 0.01) lower compared to natural light conditions. The Ant content was relatively low in P. amurensis Maxim, and it remained same under different light intensities (Fig. 3C). Under low light conditions, the Ant/Chl ratio in P. amurensis Maxim was 50.23% (p < 0.05) higher compared to natural light, which appeared to be caused by the high production of chlorophyll in the leaves of P. amurensis Maxim under low light condition. By contrast, the Ant/Chl ratio in P. opulifolius “Diabolo” under low light condition was 44.33% (p < 0.01) lower compared to the natural light.

Figure 3 Content of Car (A), Car/Chl (B), Content of Ant (C) and Ant/Chl (D) in leaves of P. amurensis Maxim and P. opulifolius “Diabolo” under natural and low light conditions.

CaBar graphs depict mean ± SE, values followed by different small letters mean significant difference (p < 0.05), values followed by different capital letters mean very significant difference (p < 0.01).

Specific leaf area (SLA) of P. amurensis Maxim and P. opulifolius “Diabolo” under different light intensities

Under natural light, the SLA of P. amurensis Maxim was 15.03% (p > 0.05) higher than that of P. opulifolius “Diabolo,” yet the difference was not statistically significant. The SLA was 58.79% (p < 0.05) higher in P. amurensis Maxim and 55.17% (p < 0.05) higher P. opulifolius “Diabolo” under low light compared to natural light conditions. These results indicate that the leaves of these two cultivars became thinner in response to the low light. Under low light, the SLA of P. amurensis Maxim was 17.72% (p > 0.05) higher than that of P. opulifolius “Diabolo,” but the difference was not statistically significant (Fig. 4).

Figure 4 Difference in specific leaf area (SLA) in leaves of P. amurensis Maxim and P. opulifolius “Diabolo” under different light intensities.

Bar graphs depict mean ± SE, values followed by different small letters mean significant difference (p < 0.05), values followed by different capital letters mean very significant difference (p < 0.01).

Light responsive curve of P. amurensis Maxim and P. opulifolius “Diabolo” under different light intensities

The changes in the leaves as we’ve observed above suggest that the cultivars would experience a change in how the leaves respond to light. To test this, we measured Pn of both P. amurensis Maxim and P. opulifolius “Diabolo,” and observed an increase in Pn along with increasing PFD (Fig. 5). There was an obvious saturation of Pn in both plants under different illumination intensities, and the Pn under lower PFD was significantly reduced in both cultivars under low light compared to under natural light. The regression analysis of the Pn-PDF curve revealed that Apparent quantum yield (AQY), Pmax and the light saturation point (LSP) in P. amurensis Maxim were 3.84, 16.88 and 5.14% higher than that in P. opulifolius “Diabolo” under natural light respectively. The light compensation point (LCP) in P. amurensis Maxim was 7.73% lower than that in P. opulifolius “Diabolo,” indicating that the photosynthetic capacity in the leaves of P. amurensis Maxim was greater than that of P. opulifolius “Diabolo” (Table 1). Under low light, AQY, Pmax, LCP and LSP in the leaves of both Physocarpus cultivars were significantly reduced in response to low light, however, these parameters were reduced to a lesser extent in P. opulifolius “Diabolo.” In addition, there was an increase of 29.42% in AQY, 42.39% in Pmax and 18.37% in LSP in P. opulifolius “Diabolo” under low light conditions when compared to P. amurensis Maxim. These results suggest that, the photosynthetic capacity was greater in the leaves of P. opulifolius “Diabolo” than in the leaves of P. amurensis Maxim under low light conditions, which was opposite to the results we observed in plants under natural light.

Figure 5 Net photosynthesis rate in response to illumination intensity in leaves of P. amurensis Maxim and P. opulifolius “Diabolo” under different light intensities.

Black squares denote.., white squares denote P. opulifolius under natural light conditions, and white triangles denote P. opulifolius under low light conditions. Error bars depict X.

Table 1 Photosynthesis parameters in leaves of P. amurensis Maxim and P. opulifolius “Diabolo” under different light intensities.

Parameters	P. amurensis	P. opulifolius	P. amurensis under low light	P. opulifolius under low light	
AQY	0.056	0.052	0.034	0.044	
Pmax (μmol · m−2 · s−1)	12.53	10.72	4.76	6.65	
LCP (μmol · m−2 · s−1)	25.89	27.88	20.07	20.00	
LSP (μmol · m−2 · s−1)	538.67	512.35	351.03	415.52	

OJIP curve of P. amurensis Maxim and P. opulifolius “Diabolo” under different light intensities

The relative fluorescence intensity (Ft) in the leaves of P. opulifolius “Diabolo” was lower than that of P. amurensis Maxim at all time points under natural light (Fig. 6A). The relative fluorescence intensity at time 0 (Fo) in P. amurensis Maxim under low light condition was significantly higher compared to natural light, whereas the relative fluorescence intensity at time P (Fm) was lower, resulting in a flatter OJIP curve in P. amurensis Maxim in response to low light (Figs. 6B–6E). On the contrary, Ft in the leaves of P. opulifolius “Diabolo” under low light condition was significantly higher, approaching the level of Ft in P. amurensis Maxim. Fo and Fm were significantly higher in the leaves of P. amurensis Maxim under low light compared to natural light (p < 0.05) (Figs. 6B–6E), whereas the different in FJ and FI were not statistically significant (p > 0.05) (Figs. 6C and 6D). By contrast, all Ft values in the leaves of P. opulifolius “Diabolo” were markedly higher in low light compared to natural light conditions (p < 0.05).

Figure 6 (A) is Chlorophyll α fluorescence transients in leaves of two cultivars of Physocarpus under different light intensities.

P. amurensis in natural light is depicted in dark green, and low light in light green. P. opulifolius is depicted in dark purple triangles for natural light, and light purple triangles for low light conditions. Fo (B), FJ (C), FI (D) and Fm (E) in leaves of two cultivars of Physocarpus under different light intensities. Bar graphs depict mean and SE, values followed by different small letters mean significant difference (p < 0.05).

Photochemical activity of PS II in the leaves of P. amurensis Maxim and P. opulifolius “Diabolo” under different light intensities

Fv/Fm and PIABS values were significantly lower in the leaves of both cultivars of Physocarpus under low light (Figs. 7A and 7B). Specifically, Fv/Fm was decreased by 17.31% (p < 0.05) P. amurensis Maxim under low light condition compared to natural light, while Fv/Fm was only 2.84% (p > 0.05) lower in P. opulifolius “Diabolo” under low light condition. The changes in PIABS in the leaves of both Physocarpus were markedly reduced in response to low light, with a decreased of 59.40% (p < 0.05) in P. amurensis Maxim and 48.13% (p < 0.05) in P. opulifolius “Diabolo,” compared to natural light. Moreover, P. opulifolius “Diabolo” had 19.51% (p < 0.05) higher Fv/Fm and 169.11% (p < 0.05) higher PIABS values compared to P. amurensis Maxim.

Figure 7 Fv/Fm (A) and PIABS (B) in leaves of two cultivars Physocarpus under different light intensities.

Bar graph depicts mean and SE, values followed by different small letters mean significant difference (p < 0.05).

Standard OJIP curve in the leaves of P. amurensis Maxim and P. opulifolius “Diabolo” under different light intensities

We next standardized the OJIP curves by defining Fo as 0, and Fm as 1 (Fig. 8A). The relative variable fluorescence (Vt) under low light condition was higher at all time points in both cultivars of Physocarpus compared to natural conditions. The degree of change in the relative variable fluorescence at time point I (VI) was greater than that at time point J (VJ). P. amurensis Maxim had significantly higher VI and VJ than P. opulifolius “Diabolo” under natural light. In response to low light, VJ in P. amurensis Maxim and P. opulifolius “Diabolo” was increased by 12.43% (p < 0.05) and 18.41% (p < 0.05), respectively, while VI displayed a higher difference of 25.22% (p < 0.05) and 43.80% (p < 0.05), respectively (Figs. 8B and 8C).

Figure 8 (A) is Chlorophyll α fluorescence transients in leaves of 2 cultivars Physocarpus under different light intensities.

The rise kinetics of relative variable fluorescence Vt = (Ft − Fo)/(Fm − Fo) and difference of VJ (B) and VI (C) in leaves of 2 cultivars Physocarpus under different light intensities. Bar graph depicts mean and SE, values followed by different small letters mean significant difference (p < 0.05).

Standard OJ and OK curves in the leaves of two cultivars of Physocarpus under different light intensities

The OJIP curves were standardized by O-J and O-K. As shown in Figs. 9A and 9B, the relative variable fluorescence at 0.3 ms (time point K) on the standardized O-J curve (VK) and the relative variable fluorescence at 0.15 ms (time point L) on the standardized O-K curve (VL) were significantly different in low light compared to natural light conditions. VK in the leaves of P. amurensis Maxim and P. opulifolius “Diabolo” under low light condition was 8.61% (p > 0.05) and 14.78% (p > 0.05) higher, respectively, and VL in these two cultivars of Physocarpus under low light was increased by 10.84% (p > 0.05) and 3.73% (p > 0.05), respectively. The changes of VK and VL were not statistically significant (Figs. 9C and 9D).

Figure 9 The rise kinetics of relative variable fluorescence Vt = (Ft − Fo)/(Fm − Fo) in leaves of 2 cultivars Physocarpus under different light intensities in 0.3 ms (A) and 0.15 ms (B). The rise kinetics of relative variable fluorescence Vt = (Ft − Fo)/(Fm − Fo) and difference of VK (C) and VL (C) in leaves of 2 cultivars Physocarpus under different light intensities.

Bar graph depicts mean and SE, values followed by different small letters mean significant difference (p < 0.05).

Parameters of energy distribution in the leaves of Physocarpus under different light intensities

Ψo and φEo in both cultivars of Physocarpus were significantly reduced in response to low light, while φDo was increased (Figs. 10A–10C). Ψo was 7.30% (p < 0.05) lower in P. amurensis Maxim and 7.86% (p < 0.05) lower in P. opulifolius “Diabolo.” By contrast, φEo was reduced by 22.89% (p < 0.05) in P. amurensis Maxim and 10.57% (p < 0.05) in P. opulifolius “Diabolo” when exposed to low light, while φDo increased by 73.64% (p < 0.05) and 13.27% (p < 0.05), respectively, with a greater degree of change in P. amurensis Maxim.

Figure 10 Difference of energy flux ratios in leaves of two cultivars of Physocarpus under different light intensities, Ψo (A), ϕEo (B) and ϕDo (C).

Bar graph depicts mean and SE, values followed by different small letters mean significant difference (p < 0.05).

Parameters of energy flux per reaction center in the leaves of two cultivars of Physocarpus under different light intensities

The absorption of luminous energy (ABS/RC) per reaction center was significantly increased in both cultivars of Physocarpus in response to low light conditions, which led to increased values of all parameters of energy flux per reaction center (Fig. 11A). However, ETo/RC in the leaves of Physocarpus did not differ significantly between natural light and low light (Fig. 11C). TRo/RC and DIo/RC in the leaves of P. amurensis Maxim under low light condition were 9.1% (p < 0.05) and 150.54% (p < 0.05) higher, respectively. TRo/RC and DIo/RC in the leaves of P. opulifolius “Diabolo” under low light condition were increased by 15.81% (p < 0.05) and 34.68% (p < 0.05), respectively. Hence, DIo/RC increased to a greater degree in P. amurensis Maxim compared to P. opulifolius “Diabolo” under low light conditions.

Figure 11 Difference of specific fluxes per reaction center in leaves of 2 cultivars Physocarpus under different light intensities, ABS/RC (A), TRo/RC (B), ETo/RC (C) and DIo/RC (D).

Bar graph depicts mean and SE, values followed by different small letters mean significant difference (p < 0.05).

Discussion

The mesophyll cells of higher plants contain a variety of pigments including chlorophyll, carotenoid and anthocyanin. The relative amount and location of these pigments in the mesophyll cells determines the color and photosynthetic function of the leaf. The leaf turns green when chlorophyll is abundant, and turns yellow or orange when carotenoid is dominantly present. When the amount of anthocyanin is abundant, the original green color of the leaf will be concealed by purple, fuchsia or red. Interestingly, the amount of various pigments can change under different light intensities. Under low light, plants often gain features to capture of more optical energy, such as enlargement of leaf area and increase of chlorophyll content (Johnson et al., 2005). In contrast, the plant will augment the synthesis of lutein in response to strong light intensity in order to protect normal function of PS by dissipating the excessive optical energy through xanthophyll cycling. In some plants, especially in new leaves with underdeveloped PS, the synthesis of anthocyanin is enhanced under a strong light to filter and attenuate the high light intensity in order to protect mesophyll cells (Lebkuecher et al., 1999). In the present study, we observed a significant increase in SLA in both cultivars of Physocarpus under low light, likely for better absorption of the optical energy. Moreover, the contents of chlorophyll were markedly elevated in both Physocarpus in response to low light. Notably, the extent of the increase was higher for Chlb than Chla, resulting in a reduced Chla/b ratio in response to low light, indicating that the increase of chlorophyll content in response to low light was mainly attributed to the enhanced production of Chlb. It has been proposed that Chla is the “converter” and the reaction center of optical energy in addition to absorbing light, whereas Chlb only functions in optical energy absorption (Zhang et al., 2013). Hence, in the presence of sufficient reaction centers, the two cultivars of Physocarpus augmented the synthesis of Chlb that does not exhibit the property of reaction center, in order to capture more light under a low light intensity. This may be a more “economic” strategy in adaption of low light intensity. In addition, the increased amount of Chlb could also help with the absorption of blue-violet light under low light, and this is an adaptive mechanism to low light to improve growth of the plants. The increase of Chlb in the two cultivars of Physocarpus under low light is consistent with the observations in Ligustrum robustum by Yan and in Ardisia violacea by Zhang in the studies on the adaptive responses to low light intensity (Yan & Wang, 2013; Zhang et al., 2014). We found some studies have shown that the key enzymes chlorophylla oxygenase (CAO) in the process of plant chlorophyll b (chl b) synthesis excess expression can increase the LHCII protein expression, the proportion of the PSI and PSII (Tanaka et al., 1998). In plant thylakoid, chlorophyll b (chl b) related proteins mainly adhesion in PSII (Horton, Ruban & Walters, 1996). In this study, Physocarpus amurensis Maxim and Physocarpus opulifolius “Diabolo” leaf chlorophyll b (chl b) content increased significantly under the low light, and chlorophyll a (chl a) changed non-significantly, which can result in two kinds of experimental materials energy captured on PSII, adjust the excitation energy distribution between PSI and PSII under the low light (Vink et al., 2004; Depége, Bellafiore & Rochaix, 2003). Comparing the two cultivars of Physocarpus, the content of chlorophyll in the leaves of P. opulifolius “Diabolo” was higher than that of P. amurensis Maxim under both natural and low light intensities. However, due to the great abundance of anthocyanin in the leaves of P. opulifolius “Diabolo” under natural light, the green color of the leaves was concealed by the purple pigments. Under low light condition, the amount of carotenoids was slightly increased in the leaves of both Physocarpus, while the amount of anthocyanin was dramatically reduced in the leaves of P. opulifolius “Diabolo,” resulting in a significant reduction of the Ant/Chl ratio. Hence, in addition to the increased production of chlorophyll, the decreased anthocyanin content also directly contributed to the reduced Ant/Chl ratio in the leaves of P. opulifolius “Diabolo” in response to low light. Collectively, we observed that both P. amurensis Maxim and P. opulifolius “Diabolo” promoted light absorption under a low light intensity by increasing SLA and the content of chlorophyll in the leaves. Meanwhile, P. opulifolius “Diabolo” also showed decreased content of anthocyanin, likely to reduce the shielding of light energy to capture more light.

As one of the most light-sensitive components in plants, the variety and amount of pigments in leaves not only affect light absorption, but also directly interferes with a series of physiological processes during photosynthesis (Lu et al., 2003; Abadía et al., 1999). In this study, the changes in the contents of pigments in the leaves of both cultivars of Physocarpus in response to low light led to alteration in the photosynthetic function. Our results showed that, although the chlorophyll content was higher in the leaves of P. opulifolius “Diabolo” under natural light, the photosynthetic capacity in the leaves of P. opulifolius “Diabolo” was lower than P. amurensis Maxim. It is probably because of the optical filtration effect by the high amount of anthocyanin in the leaves of P. opulifolius “Diabolo” under natural light (Chalker-Scott, 1999), thus the actual light absorption was lower in the leaves of P. opulifolius “Diabolo.” In addition, AQY, Pmax, LCP and LSP were significantly reduced in response to low light in both cultivars of Physocarpus, implying that the photosynthetic activity was inhibited, and light absorption and utilization was dampened. The utilization of high-intensity light, in particular, was reduced significantly. By contrast, the reduction of LCP indicated that the utilization of low-intensity light was enhanced in both Physocarpus in low light conditions. These observations are consistent with the physiological features of the plants under a low light. Furthermore, in response to low light, all photosynthetic parameters were decreased to a less extent in P. opulifolius “Diabolo” than that in P. amurensis Maxim, implying that the photosynthetic capacity in P. opulifolius “Diabolo” was higher under low light. This is likely to be associated with the higher amount of chlorophyll, and the smaller SLA in P. opulifolius “Diabolo” under low light, and suggests that the leaves of P. opulifolius “Diabolo” was relatively thicker under low light and favored photosynthesis.

Despite the reduction in light absorption by the leaves, low light can directly interfere with the utilization of optical energy and alter the primary photochemical reaction in photosynthesis. The chlorophyll fluorescence parameters can provide information about primary photochemical reactions in the leaves. Fast chlorophyll florescence dynamics can indicate the structure and function of the PS II reaction center during the process of chlorophyll fluorescence quenching, therefore is widely used to study the physiological response and the adaption mechanisms to stress in plants (Lu et al., 2003). In the current study, Fv/Fm and PIABS in the leaves of both cultivars of Physocarpus were significantly decreased in response to low light compared to natural light, and PIABS was decreased to a greater extent than Fv/Fm. These results suggest that the structure and function of the PS II reaction complex were altered by low light in the leaves of the two cultivars of Physocarpus, resulting in a reduced quantum efficiency of the primary photochemical reaction in PS II and a reduced activity of PS II reaction center. In addition, PIABS was more sensitive to low light than Fv/Fm in both Physocarpus, suggesting that PIABS can better reflect the PS II function in the leaves of both Physocarpus under low light condition as compared to Fv/Fm. This is consistent with the stress responses in most plants (Wen et al., 2005). On the other hand, Fv/Fm and PIABS under low light were higher in the leaves of P. opulifolius “Diabolo” than that of P. amurensis Maxim, suggesting that the PS II reaction center in P. opulifolius “Diabolo” was more active under low light, and the primary photochemical reaction in P. opulifolius “Diabolo” was less affected by low light.

The OJIP curves showed that the relative fluorescence intensity (Ft) in the leaves of P. opulifolius “Diabolo” was lower than that of P. amurensis Maxim at all time points. This is probably because the great amount of anthocyanin in the leaves of P. opulifolius “Diabolo” dampening the light intensity. In response to low light, however, the synthesis of anthocyanin in the leaves of P. opulifolius “Diabolo” was inhibited, turning leaves from purple to green, resulting in a significantly increased Ft in the OJIP curve at all points. This observation further corroborates with the fact that the presence of anthocyanin can reduce the fluorescence quenching in the leaves to a certain degree, suggesting that anthocyanin may exhibit a photoprotective property (Wang et al., 2012; Zeliou, Manetas & Petropoulou, 2009). The original OJIP curve was largely affected by the environment. Hence, in order to analyze the fluorescent changes at specific points, the OJIP curves are often standardized, such that all OJIP curves share a common starting point and a common end point. Under low light, the standardized OJIP curves of both Physocarpus showed an increase in the relative variable fluorescence at point J (VJ) and point I (VI), indicating an accumulative quantity of equation QA− (Strasser, Srivastava & Govindjee, 1995; Govindjee, 1995). Previous studies by Schansker, Tóth & Strasser (2005) and Li et al. (2009), demonstrated that increasing VJ can reduce the oxidation of plastoquinone QB, and higher VI can reduce the reoxidation of PQH2. This suggests that the increase of VJ and VI resulted from the blockade of electron transfer from the primary electron acceptor QA− to the secondary electron acceptor QB, as well as from QB to PQ in the PS II reaction complex. In our study, VI increased to a greater degree than VJ in the leaves of both Physocarpus in response to low light, suggesting that the reason for the blockage of electron transfer on the electron acceptor side of PS II in the leaves of both Physocarpus under low light was related to the reduced capacity of electron acceptance of QB and PQ, whereby the reduced storage capacity of PQ served as the major rate-limiting step. We found that VJ on the standardized OJIP curve of both cultivars of Physocarpus under low light was significantly increased, indicating a massive accumulation of QA−. In order to eliminate the effect of the electron acceptor side of PS II, the O-J and O-K curves were standardized. The results showed that the relative variable fluorescence at 0.3 ms (time point K) on the standardized O-J curve (VK) and the relative variable fluorescence at 0.15 ms (time point L) on the standardized O-K curve (VL) were higher in both Physocarpus cultivars under low light. It has been reported that the increasing of VK may be affected by the state of OEC and the link between PS II units (Jiang et al., 2006; Strasser, Srivastava & Govindjee, 1995), and the increase of VK is mainly associated with the inhibition of activity of PS II electron donor side, especially the OEC. Bertamini and Nedunchezhian found that the donor side of PS II was more susceptible to inhibition under stress primarily as a result of the reduction of the 33 kDa hydrolyzed compound protein (Bertamini & Nedunchezhian, 2003a; Bertamini & Nedunchezhian, 2003b). The activity of OEC is often inhibited under stress, leading to the blockade of electron transfer from the electron donor side to the electron acceptor side (Bertamini & Nedunchezhian, 2003a). Moreover, the increase of VL is mainly associated with the damage of thylakoid membrane and the dissociation of thylakoids in the chloroplasts (Ye et al., 2013). In the present study, we observed an increase in VK and VL in both Physocarpus cultivars under low light, but the differences were not significant, suggesting that the low light intensity led to the reduction in the activity of OEC on the electron donor side and the peroxidation of thylakoid membrane in the leaves of both Physocarpus, but did not result in the inactivation of OEC or evident dissociation of thylakoids. Yet this speculation needs to be verified in further studies.

The low light intensity altered light utilization by the PS II reaction centers and the energy flux per PS II reaction center in the leaves of both Physocarpus. Specifically, in response to low light, the quantum yield of the absorbed light energy used for electron transfer (φEo) in the PS II reaction center was reduced, whereas the quantum yield of the absorbed light energy used for dissipation (φDo) was increased. Moreover, the energy flux parameters per PSII reaction center, including ABS/RC, TRo/RC, ETo/RC and DIo/RC, all showed an increasing trend in the leaves of both Physocarpus cultivars under low light. A decrease of φEo and an increase of ETo/RC indicated that there was higher utilization of absorbed light energy on electron transfer in a single PS II reaction center, although there was lower quantum yield of the absorbed light energy used for electron transfer in all PS II reaction centers. This is indirect evidence for the reduction in the number of active PS II reaction centers in the leaves of both Physocarpus cultivars in response to low light. Studies have found that in the situation that a number of PS II reaction centers were inactivated under stress, the function of the antenna pigments in the remaining active reaction centers increases in order to guarantee energy supply (Demetriou et al., 2007). Our study showed similar results, that ABS/RC in the leaves of both Physocarpus was significantly increased in response to low light, which may be due to the enhanced activity of the antenna pigments per reaction center as an adaptive mechanism to the low light intensity. The energy flux to reduction of QA pre reaction center (TRo/RC) was markedly increased in response to low light, suggesting that the electron transfer from pheophytin (Pheo) to QA (i.e., reduction of QA to QA−) during photosynthesis in the leaves of both Physocarpus was minimally affected by the low light intensity. However, the ratio of energy flux to the electron transfer downstream of QA− (Ψo) in the PS II reaction centers was significantly decreased in low light, further demonstrating the blockade of electron transfer on the electron acceptor side of PS II occurred after QA− in the leaves of both Physocarpus under low light. By contrast, low light conditions had little effect on the electron transfer from Pheo to QA, indicating that low light resulted in an accumulation of QA− in the electron transport chain. This is consistent with the results of VJ and VI increasing as described above. In short, the PS II electron acceptor side appears to be the target site of the effect of low light on plants. In addition, φDo and DIo/RC increased in the leaves of both Physocarpus cultivars under low light, indicating that low light led to reduced utilization of optical energy in the leaves of both Physocarpus cultivars, and that the optical energy was dissipated mainly via heat and fluorescence when the activity of PS II reaction centers was dampened. Moreover, the extent of the decrease of φEo and the increases of φDo and DIo/RC was significantly less in P. opulifolius “Diabolo” than that in P. amurensis Maxim, suggesting that the utilization of optical energy and the activity of PS II reaction centers were higher in P. opulifolius “Diabolo.” Thus, the function of PS II in P. amurensis Maxim was more sensitive to light than that in P. opulifolius “Diabolo.”

Conclusion

Although the content of chlorophyll in the leaves of P. opulifolius “Diabolo” is higher than that of P. amurensis Maxim under natural light, P. opulifolius “Diabolo” has purple leaves caused by the presence of high anthocyanin in the leaves. Moreover, the presence of anthocyanin reduces the capacity of photosynthetic carbon assimilation in P. opulifolius “Diabolo.” In response to low light, the content of chlorophyll in the leaves of both P. amurensis Maxim and P. opulifolius “Diabolo” increased, yet the capacity of photosynthetic carbon assimilation and the photochemical activity of PS II are significantly reduced. The relatively higher chlorophyll content and the smaller SLA in the leaves of P. opulifolius “Diabolo” provided this cultivar with a greater photosynthetic capacity as compared to P. amurensis Maxim. Our study indicates that the ornamental value of P. opulifolius “Diabolo” is higher than that of P. amurensis Maxim under natural light, and under low light, the effect of growth inhibition was lower in P. opulifolius “Diabolo.” Hence, the imported P. opulifolius “Diabolo” exhibits an advantage in shade tolerance, but the ornamental value of P. opulifolius “Diabolo” is lower under shaded conditions. Under a low light intensity, P. opulifolius “Diabolo” maintains a relatively high activity of primary photochemical reaction in PS II by altering the composition of photosynthetic pigments in the leaves, which is an important mechanism for its better shade-tolerant property over P. amurensis Maxim.

Supplemental Information

Supplemental Information 1 Raw Data.

Click here for additional data file.

Additional Information and Declarations

Competing Interests

Author Contributions

Data Deposition

The authors declare that they have no competing interests.

Huihui Zhang conceived and designed the experiments, performed the experiments, analyzed the data, contributed reagents/materials/analysis tools, wrote the paper, prepared figures and/or tables.

Haixiu Zhong performed the experiments, prepared figures and/or tables.

JIfeng Wang performed the experiments, prepared figures and/or tables.

Xin Sui analyzed the data.

Nan Xu conceived and designed the experiments, contributed reagents/materials/analysis tools, reviewed drafts of the paper.

The following information was supplied regarding data availability:

The raw data has been supplied as Supplemental Dataset Files.

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
