# Peer review of "Adaptive changes in chlorophyll content and photosynthetic features to low light in Physocarpus amurensis Maxim and Physocarpus opulifolius “Diabolo”"

_PeerJ, doi:10.7717/peerj.2125_

## Round 0.1 · original submission · Minor Revisions

1. Please carefully check the concern raised by the reviewer 3 about "the effect of change in anthocyanin content on effective photosynthetic light". The authors should explain or rationalize this issue in the discussion section, and correspondingly revise some data interpretation if this is necessary;

2. The authors also need to pay attention to some errors and terminology in the text as reviewer 1 and 2 pointed out, make revisions;

3. Add more discussion point as suggested by reviewer 2 and pay attention to some data under-interpreted (Chla/b ration – antennae size).

Reviewer 1 ·

Basic reporting

The manuscript is well written based upon the experimental data presented and in accordance with the guideline and template set by PeerJ. It should get published.

Experimental design

The study was well designed and defined with clear and meaningful research questions. The research was conducted in a good standard in this field.

Validity of the findings

No comments.

Additional comments

There are a number of minor issues listed below, which need to be revised by the authors before accepted.
Line 78, 81, 82 and 83: Seedlings and cuttings are two different terms used in scientific field. Seedlings are the young plants produced from seeds while cuttings are usually used for that which is vegetatively produced. The authors need to clearify which materials were used, cuttings or seedlings?
Line 88-89: Need to address what device/instrument is used to measure the light intesity and what source of the artificial light used.
Line 93: Figure 1 should be in result section instead of in materials and methods section.
Line 104, 181, 197, 200, 201, and 203: Ant? SLA?PFD?AQY?LCP?What are they? Any abbreviation you use first time in the manuscript should have a full name.
Line 167 and 172: Car/Chl and Ant/Chl should be Car/Chl(a+b) and Ant/Chl(a+b)?
Line 48,51,199, 407, 418, 446: spelling errors, liu?PDF?(36)?PSS?and prensence?

Reviewer 2 ·

Basic reporting

The article compares the photosynthetic behaviour of two Physocarpus cultivars under low-light conditions and try to explain the differences on the basis of pigment variations and the photosynthetic electron transport yield.
In general the article is written in a clear English although some sentences should be revised. The introduction and background is enough. The article is well structured and Figures and Tables are clear. The conclusions are appropriate but authors should add some discussion concerning the role of the PSII/PSI balance in the shade-tolerance of "Diabolo" cultivar (see comments for the authors).

Experimental design

The experimental design is correct.

Validity of the findings

The data are statistically controlled. The conclusions are well argumented, but some additional discussion should be introduced (see comments for the authors).

Additional comments

Minor comments:

Line 105. Write Anthocyanin (Ant)
Line 108. 350μmol·m-2·s-1, separate with a blank space.
Line 110. 0μmol·m-2·s-1, separate with a blank space.
Line 113. 350μmol·m-2·s-1CO2) under1200μmol·m-2·s-1 PFD, separate with a blank space.
Line 115. Revise the grammar in the sentence “Having been dark….…” Start with a subject.
Line 173. Revise English grammar
Line 199. Revise the English grammar
Lines 328 - 341. The increase of Chl b over Chla is due to an increase of the ratio PSII / PSI. Consequently, this fact increases the grana formation in thylakoid membranes where the PSII is mainly localized. The composition of Chla and Chlb of both PSII and PSI antenna complexes is different. This point should be discussed. Authors should also discuss the variations of the balance between PSII and PSI in both cultivars in the context of the shade-tolerance.

Is it possible that the gene regulation of PSII components is favoured under low light in P. opulifolius “Diabolo” compared with P. amurensis Maxim?

Reviewer 3 ·

Basic reporting

Some figures are not propperly described, e.g. Fig. 5, how was it normalized.
In conclusion, as the effect of change in anthocyanin content on effective photosynthetic light is not take into account properly, the manuscript need to be improved and data re-interpreted based on the anthocyanin effect on photosynthetically effective light. Therefore, I would not accept the manuscript at this stage. I have also feeling that some data are overinterpreted (OJIP parameters) and some are not used for interpretation (Chla/b ration – antennae size)

Experimental design

The point is, authors have to take into account the fact that anthocyanin is not photosynthetically active pigment and it acts as a screening flavonoid that reduces effective light for photosynthesis. Therefore, all photosynthetic measurements they have done (photosynthetic rate, fluorescence etc.) are affected by a difference in “effective light” intensity (non-absorbed by anthocyanin). This analysis is necessary (at least a deconvolution of spectra measured with intact leaves….) and crucial for any other analysis.

Validity of the findings

The manuscript compares adaptive changes in two Psychocarus cultivars (native and imported) to change in light conditions (natural and 100uE). There is a striking difference between these species, the North American cultivar adapt to natural condition by an increase in anthocyanin content (clearly visible from the pictures and chemical analysis). Well, considering the fact, what I really miss is a comparison of at absorption spectra of 4 used variants of leaves (NL, LL x2 cultivatrs).

Additional comments

The manuscript compares adaptive changes in two Psychocarus cultivars (native and imported) to change in light conditions (natural and 100uE). There is a striking difference between these species, the North American cultivar adapt to natural condition by an increase in anthocyanin content (clearly visible from the pictures and chemical analysis). Well, considering the fact, what I really miss is a comparison of at absorption spectra of 4 used variants of leaves (NL, LL x2 cultivatrs).
The point is, authors have to take into account the fact that anthocyanin is not photosynthetically active pigment and it acts as a screening flavonoid that reduces effective light for photosynthesis. Therefore, all photosynthetic measurements they have done (photosynthetic rate, fluorescence etc.) are affected by a difference in “effective light” intensity (non-absorbed by anthocyanin). This analysis is necessary (at least a deconvolution of spectra measured with intact leaves….) and crucial for any other analysis.

Other more details comments:
- Some data are not properly described (gas exchange parameters – how did you normalize it? To chltophylls, did you take into account difference in anthocyanin? – It need to be added!
- I would not use so many parameters taken from OJIP kinetics, it is so confusing and does not say too much about situation on light (in comparison to quenching analysis) – It is a pitty that you did not measure photochemical and non-photochemical quenching on actinic light – those are more usefull parameters
- Chla/chlb ratio tells you about amount of antennae per RCII, discuss data in the sence…

In conclusion, as the effect of change in anthocyanin content on effective photosynthetic light is not take into account properly, the manuscript need to be improved and data re-interpreted based on the anthocyanin effect on photosynthetically effective light. Therefore, I would not accept the manuscript at this stage. I have also feeling that some data are overinterpreted (OJIP parameters) and some are not used for interpretation (Chla/b ration – antennae size)

---

## Round 0.2 · accepted · Accept

Thanks you for your submission, which is now accepted.

Reviewer 1 ·

Basic reporting

no comments.

Experimental design

No comments.

Validity of the findings

No comments.